# mRNA in the Context of Protein Replacement Therapy

**DOI:** 10.3390/pharmaceutics15010166

**Published:** 2023-01-03

**Authors:** Theofanis Vavilis, Eleni Stamoula, Alexandra Ainatzoglou, Athanasios Sachinidis, Malamatenia Lamprinou, Ioannis Dardalas, Ioannis S. Vizirianakis

**Affiliations:** 1Laboratory of Biology and Genetics, School of Medicine, Aristotle University of Thessaloniki, 54124 Thessaloniki, Greece; 2Department of Dentistry, European University Cyprus, Nicosia 2404, Cyprus; 3Centre of Systems Biology, Department of Biotechnology, Biomedical Research Foundation of the Academy of Athens, 11527 Athens, Greece; 4Department of Clinical Pharmacology, School of Medicine, Aristotle University of Thessaloniki, 54124 Thessaloniki, Greece; 54th Department of Internal Medicine, Hippokration General Hospital, School of Medicine, Aristotle University of Thessaloniki, 54642 Thessaloniki, Greece; 6Laboratory of Pharmacology, School of Pharmacy, Aristotle University of Thessaloniki, 54124 Thessaloniki, Greece; 7Department of Life & Health Sciences, School of Sciences and Engineering, University of Nicosia, Nicosia 1700, Cyprus

**Keywords:** mRNA, protein replacement therapy, modRNA, lipid nanoparticles, nanomedicine, metabolic diseases, hepatic diseases, cardiovascular diseases, lung diseases, hematologic diseases

## Abstract

Protein replacement therapy is an umbrella term used for medical treatments that aim to substitute or replenish specific protein deficiencies that result either from the protein being absent or non-functional due to mutations in affected patients. Traditionally, such an approach requires a well characterized but arduous and expensive protein production procedure that employs in vitro expression and translation of the pharmaceutical protein in host cells, followed by extensive purification steps. In the wake of the SARS-CoV-2 pandemic, mRNA-based pharmaceuticals were recruited to achieve rapid in vivo production of antigens, proving that the in vivo translation of exogenously administered mRNA is nowadays a viable therapeutic option. In addition, the urgency of the situation and worldwide demand for mRNA-based medicine has led to an evolution in relevant technologies, such as in vitro transcription and nanolipid carriers. In this review, we present preclinical and clinical applications of mRNA as a tool for protein replacement therapy, alongside with information pertaining to the manufacture of modified mRNA through in vitro transcription, carriers employed for its intracellular delivery and critical quality attributes pertaining to the finished product.

## 1. Introduction

Conventional therapies that address diseases arising from genetic defects and inborn errors of metabolism generally tend to treat the symptom rather than the cause. Hence, therapeutics that aim at addressing the root of the problem, namely DNA and more recently RNA therapies, have been long been sought after by the scientific community. While gene therapy, in the form of gene editing, is a commonly discussed method, other equally promising approaches such as RNA therapeutics have emerged, which could have advantages compared to more invasive or permanent procedures. RNA therapeutics have been based on two principal functions of the RNA molecule, namely its regulatory and encoding capacities. On one hand, its regulatory functions have been applied using antisense oligonucleotides (ASOs) that bind to complementary sequences of host RNA transcripts, modulating their expression. Currently, there already exist FDA-approved ASO drugs for diseases such as Duchenne muscular dystrophy, familial amyloid polyneuropathy, spinal muscular atrophy and familial chylomicronemia syndrome [1,2,3,4]. Similarly, other molecules employed to exert the regulatory functions of RNA are small interfering RNAs (siRNAs) and microRNAs. The former act through their interplay with the RNA-induced silencing complex (RISC), leading to RNA interference (RNAi), meaning that a siRNA has the potential to target a prespecified mRNA molecule and induce its degradation, completely blocking the expression of a certain gene [5]. Drugs based on siRNAs have already been launched to treat TTR-amyloidosis and hepatic porphyria [6,7]. Regarding microRNAs, these molecules inhibit target mRNAs by forming a complex with RISC and downstream suppressing mRNA translation or contributing to its degradation or cleavage. Belonging to this regulatory concept but in a completely distinct pattern are RNA aptamers, single-stranded RNAs that interact with heterogenous targets, such as proteins and carbohydrates. This functional flexibility derives from their tertiary structure, which is more critical than their sequence [5].

On the other hand, the capacity of messenger RNA (mRNA) to encode peptides or proteins and spark their transient cellular expression has been employed to restore protein deficiencies through mRNA protein replacement therapy or to achieve antigen presentation through mRNA vaccination. It has long been established that mRNA is generated through the transcription of the genomic DNA to mediate the delivery of the genetic information to the cellular translational machinery. Soon after this discovery, there were studies putting this novel evidence to the test by injecting exogenous mRNA in vivo and monitoring its capacity to induce protein expression [8,9,10]. This was the origin of mRNA protein replacement therapy, a technique that has so far been mostly employed in studies investigating novel treatment approaches for rare monogenic diseases. These conditions are owed to single-gene defects present in the human coding genome that cause the encoded proteins to be defective or even missing. Despite their rarity, these so-called “orphan” diseases are numerous, thus collectively affecting a large portion of the global population. This fact has urged many countries to adopt legislations encouraging manufacturers to develop the respective orphan drugs in spite of their high pricing that needs to be reimbursed [11].

### Protein Replacement Options—Why Choose mRNA?

Apart from mRNA, there are also other ways to achieve protein replacement, such as gene therapy and recombinant protein production using biomolecular engineering [12]. Regarding gene therapy, its main principle is the replacement of defective or missing genes through the delivery and integration of normal genes in the affected cells, in order to fix genetic disorders responsible for disrupting key cellular pathways through the restoration of protein expression. However, the shortest way to the end goal of protein expression is through direct protein delivery to the affected tissue. This approach offers the advantage of bypassing the procedures of cellular translation, therefore facilitating dose regulation, achieving high protein levels in the tissue and offering greater control compared to gene therapy involving viral delivery. Although the administration of the desirable protein has already been successfully applied in a wide variety of diseases, warranting the substitution of hormones, enzymes, blood clotting factors and interferons, there are still numerous conditions in which the use of recombinant proteins is not applicable. This fact can be attributed to the short half-life of the administered proteins, which combined with their instability and potential immunogenicity constitute barriers against the generalizability of this approach [13]. Another crucial downside of this method is its ineligibility for the substitution of intracellular proteins, such as transcriptional factors and other regulatory molecules.

The employment of nucleic acids for the purposes of protein replacement offers greater flexibility by avoiding the challenges of direct protein delivery; however, there are also some drawbacks inherent to this approach. Regarding gene therapy, its insertion methods define both their benefits and their limitations. Although the use of viral vectors ensures the incorporation of the delivered genes into the host genome, therefore safeguarding constant expression, this can act as a double-edged sword by altering the genome and potentially promoting oncogenesis. Nonetheless, not all viral vectors are integrated into the genome, as their tropism for the tissue of interest is adequate for their selection as carriers; however, other problems may occur, such as the emergence of neutralizing antibodies against them, pharmacokinetic issues, as well as difficulties in loading the desirable gene upon them. Oppositely, non-viral delivery, as in the form of liposomes, dendrimers, solid lipid nanoparticles and polymeric micelles used to deliver plasmid DNA, has the benefit of lower immunogenic and oncogenic capacity, as well as high tissue specificity, yet displaying very poor transfection efficiency [5].

By comparatively assessing the aforementioned protein replacement approaches, one can safely draw the inference that the use of mRNA-based treatments offers significant advantages over protein or DNA-based methods. These mainly include its capacity to circumvent processes such as insertion in the nucleus and transcription, its enhanced safety due to lack of integration to the host genome and the transient nature of its effect, eliminating the possibility of mutagenic activity. From a regulatory point of view, a protein replacement therapy that utilizes mRNA does not constitute genetic alteration, and as such the development of such therapeutic approaches may be more legislatively favored compared to its DNA counterparts. However, mRNA therapy has also been associated with side-effects, as mRNA can be intracellularly cleaved by RNase, subsequently eliciting an innate immune response through the activation of Toll-like receptors (TLRs) 3, 7 and 8 present on endosomal membranes, which can eventually lead to cytokine-induced toxicity [14]. This downside was later addressed by a study reporting that the replacement of uridine by pseudouridine, with the technology of modified mRNA (modRNA), was able to curb the recognition of mRNA by TLRs and nucleases due to alterations in the secondary structure of this molecule, as discussed further on [15]. Thereafter, there have been many studies unanimously attesting to the additional benefit of modRNA in achieving higher levels of protein expression, while displaying significantly lower immunogenic effects, as observed by the downregulated activation of genes encoding interferons and the retinoic acid-inducible gene [16,17,18]. Figure 1 depicts the timeline from the discovery of mRNA up to its major introduction to the world in the form of a COVID-19 vaccine.

## 2. Considerations on mRNA Production for Protein Replacement Therapies

mRNA for protein replacement therapies can nowadays be produced in cell-free systems through the in vitro transcription reaction (IVT). The IVT takes place in a batch or continuous bioreactor that monitors and controls reaction conditions such as pH and temperature [38,39]. The main IVT reagents are the linearized DNA plasmid that plays the role of the template from which mRNA is going to be transcribed, a DNA-dependent RNA polymerase that performs the transcription, dNTPs that will serve as mRNA building blocks and in some cases a 5′ Cap analogue [40]. Furthermore, the reaction must take place in a buffered environment containing magnesium ions and sodium chloride, which calls for the use of appropriate buffers such as HEPES/Tris [41]. A cornucopia of additional reagents can be employed, which can safeguard the integrity of the reaction and the product, leading to increased final yields. Such reagents include RNAse inhibitors, pyrophosphatase, dithiothreitol and polyamines [42,43,44,45].

### mRNA Architecture for Successful Protein Production

The translational efficiency of the resulting mRNA depends on its architecture and chemical composition, which are dictated by the design of the plasmid that is subjected to IVT and the IVT reagents used in the reaction. mRNA that is suitable for protein replacement therapies must possess an appropriate 5′ Cap as well as 5′–3′ untranslated regions (UTRs) and an open reading frame engineered for maximizing protein production. The modifications employed for the optimization of mRNA (modRNA) are presented in Figure 2.

An important decision pertaining to the IVT reaction reagents for producing protein replacement therapy mRNAs is the capping method utilized. In order for mRNA to be functional, it must incorporate on its 5′ end a cap structure. Without a 5′ cap, the resulting mRNA molecule lacks a region essential to the initiation of translation and is prone to intracellular degradation [46,47]. At the time being, two options are available, namely either co-transcriptional capping or post-transcriptional capping. The former takes place during the IVT main reaction and constitutes the T7 RNA polymerase, incorporating a “Cap 1” analogue, such as the commercially available “Cleancap” by Trilink, while the mRNA is being transcribed [48,49]. The latter option utilizes an additional step, where post-IVT a 5′ cap is formed by capping enzymes of the Vaccinia virus which are further converted to cap 1 [40,50]. While this option offers a 100% capping rate compared to the 90% capping rate of the co-transcriptional method, it complicates the procedure by requiring the utilization of an additional bioreactor and purification steps [42,48,51].

The 5′ and 3′ UTR regions are areas of the mRNA that flank the amino acid coding sequence of the transcript. While not translated, they can exert a regulatory role by modulating the transcript’s interaction with RNA binding proteins and ribosomes, as well as by providing sites for microRNA binding [52,53,54,55]. An analysis of 5′ UTR regions correlated with high protein yields has shown that the design of this region on a plasmid level should utilize sequences that are devoid of start and stop codons [46]. It also appears prudent to minimize the incorporation of sequences that display increased propensity of regional secondary structure formation on the resulting mRNA, as this can in some cases hinder mRNA–ribosome interactions and diminish protein yields [56]. Regarding 3′ UTR regions, contrary to 5′ UTR, they appear to benefit from the inclusion of secondary structure-forming sequences [56]. A successful design strategy is to utilize 3′ UTR sequences of mRNAs that are abundantly expressed, such as the 3′ UTRs of α- and β-globin [36]. Other successful approaches, which can increase mRNA stability and protein yield, are the employment of multiple copies of 3′ UTRs, as highlighted by the use of two heads to tail β-globin 3′ UTRs or even the incorporation of “fusion UTRs” such as the hybrid product between Aminoterminal Enhancer of Split (AES) 3′ UTR and mitochondrially encoded 12S rRNA (mtRNR1) 3′ UTR [57,58].

The Open Reading Frame (ORF), that is the mRNA part corresponding to the final amino acid sequence of the protein, is coded in the plasmid as an intron-depleted sequence. Various modifications are employed that can increase mRNA stability, as well as make its translation more efficient, yielding more protein per mRNA molecule. At a plasmid design level, the codon sequence can be optimized by following the human codon bias, or in other words by selecting amongst synonymous codons the ones for which human tRNA species are more ample [59,60]. Such an approach can increase the translation rate, once the mRNA is in the cell. Furthermore, sequence optimization can be benefited from by modifying the sequence to ensure a high GC/AU ratio. The rationale behind this manipulation is that GC-enriched counterparts are not subjected to such an intensive post-transcriptional regulation when compared to the original AU-containing sequences; hence, they remain available to the ribosomes for translation [61]. Another method employed to deplete the U content of the transcript is the substitution of uridine by translationally tolerated analogues such as pseudouridine and N1-methylpseudourine (Ψ and m^1^Ψ, respectively) [62,63]. This substitution has the added value of conferring to the mRNA product the ability to escape recognition by intracellular Toll-like receptors (TLRs), which mediate the cell’s innate degradation response against invading mRNAs [56,64,65]. Without adequate Ψ and m^1^Ψ substitution, the supplied mRNA triggers the cellular innate immunity and is flagged as “foreign” for destruction, heavily diminishing the amount of available transcript for translation. Ψ and m^1^Ψ substitution are techniques employed at the IVT reaction level, by feeding the bioreactor with a dNTP mixture containing those analogues at the place of uridine [15].

Another feature of mRNA that adds to its stability by preventing premature degradation is the presence of a Poly(A) tail [66]. The poly-A tail is located at the 3′ end of the mature mRNA and it comprises a repeating adenine sequence. Usually, an optimized length of 100–120 adenines provides for the increased translation efficiency that is sought after in a protein replacement therapy transcript [64,67,68]. While during in vivo transcription the polyadenylation of the mRNA takes places using a separate enzyme called poly-A polymerase, an IVT employment of such a strategy would unnecessarily complicate the reaction and add to the costs [69]. IVT polymerases such as T3,T7 or SP6 can also polyadenylate the transcript, though they tend to produce end-products with a variable tail length [70]. The above complications can be side-stepped by encoding the poly-A tail in the plasmid to be transcribed, ensuring this way the uniformity of the mRNA length tail [71,72].

Following the IVT reaction, the resulting mRNA must be purified in order to remove contaminants and receive pharmaceutical-grade mRNA. Purification procedures widely employed are chromatographic procedures such as Ion Exchange, Affinity Poly(dT) and ion-pair reverse-phase chromatography [73]. These chromatographic techniques are usually paired with tangential flow diafiltration steps [74]. Figure 3 summarizes the events up to the production of pure mRNA. The product is then ready to be forwarded to the next step, which is the incorporation in the delivery vector of choice that constitutes the vehicle for cellular delivery.

## 3. Vehicles for Delivering mRNA to the Cells

Many methods of safe mRNA delivery to the target cells have been implemented, all of which have their own advantages and their own limitations. The main principles include the mRNA protection from degradation as well as its effective and prompt transportation to the target cells. Methods of transportation mainly include the viral vectors and the non-viral vectors categorized in the polymer-based vectors, the lipid-based vectors and the hybrid polymer–lipid vectors [75].

One of the most important parts of gene therapy is the effective delivery of this important gene to the target cells or target tissue. It can either replace an existing gene or a non-working gene; it may silence a gene or inactivate a gene that has been mutated to a non-favorable form [76]. Regarding gene therapy and mRNA delivery, low immunogenicity is also of vital importance [77].

Virotherapy or the utilization of genetically modified viruses in research and clinical practice has been extensively studied for decades, but it has mostly failed. A viral vector consists of a virus as a transporter of nucleic acids, possessing a protein capsid with or without the viral envelope, the gene of importance, as well as key viral elements vital for the gene integration and expression. The viral part can commonly either be a retrovirus, an adenovirus or an adeno-associated virus, but alphaviruses, picornaviruses and flavivirus have also been utilized [76]. Viral vectors usually show elevated transduction efficiency, wide-range tropism, while in addition their production can nowadays be scaled up [78]. A limitation to their success could be any pre-existing immunity to the viral vector in the target population [79].

Non-viral vectors showcase a new chapter in nanotechnology; being very promising, they have been extensively used during the COVID-19 pandemic in two mRNA vaccines currently on the market. These vectors are based on the cationic polymer nanoparticles’ (NPs) electrostatic properties, which interact with the mRNA they have encapsulated, which has a negative charge. Naturally occurring polymers include chitosan, while the synthetic ones include PLL (poly-L-lysine), PAMAM (polyamidoamine), PEI (polyethyleneimine) and PAA (polyacrylic acid) [40]. Certain properties of these NPs should be considered when choosing the optimal one, namely their charge density, their molecular weight, their toxicity profile and pharmacokinetic properties, such as distribution, biodegradation and clearance. In order to achieve the best possible results, certain modifications in these nanoparticles have to be performed [80].

Another method of mRNA efficient encapsulation and delivery involves the use of lipid nanoparticles (LNPs) or a lipid bilayer shell, consisting of cationic lipids with an aqueous center. Other molecules such as cholesterol or PEG (polyethylene glycol), in the form of PEGylated lipids, are also part of the final delivery structure [81,82,83]. A main limitation of these LNPs is their rapid clearance due to the cationic shell and its interaction with plasma proteins. Recent research and development of certain lipids such as 1,2-dioleyloxy-N,N-dimethyl-3-aminopropane (DODMA) or 1,2-dioleoyl-3-dimethylammonium propane (DODAP) achieved an almost neutral charge in circulation. As a result, the half-life of the LNPs was augmented [84]. It must be noted that critical properties of the LNPs, dictating their behavior and stability, such as particle size, delta potential (aggregation), endosomal release propensity, distribution in the body, etc., can be tailored by the careful selection of LNP constituents. The inclusion of PEG lipids is known to affect both the zeta potential and size of the resulting particle, increasing the stability of the formations by inhibiting their aggregation [85]. The percentage of PEG utilized as well as PEG conjugations also play a role in the characteristics of the LNPs [85,86]. For example, a larger PEG percentage can lead to the formation of smaller LNPs [86]. The conjugation of 1,2-dimyristoyl-rac-glycero-3-methoxy, which carries a C14 alkyl chain, can also modify behavior, providing lower circulation time in the bloodstream coupled with higher cellular delivery rates, compared to conjugates utilizing C18 1,2-distearoyl-rac-glycero-3-methoxy [87]. Cholesterol and its analogues can favorably impact the LNPs’ morphology and properties [85]. The introduction of cholesterol analogues with various modifications of the tail can change the LNP’s shape from spherical to polyhedral, displaying transfection efficiency and imbue those particles with a “homing” capability toward liver endothelial and Kupffer cells [88,89,90]. Finally, the DODMA and DODPA already mentioned, have low zeta potentials and form LNPs that are of small size and that have the propensity to accumulate in the spleen; hence, they could be of use when targeting this organ [91,92].

In the light of the SARS-CoV-2 pandemic, the production of LNPs has been streamlined to cater to the unprecedented worldwide demand for an mRNA–LNP vaccine. As such, technologies such as microfluidics have been drafted to the battlefield to reproducibly provide mRNA–LNP species with uniform qualities. Figure 4 presents a schematic representation of the most widely employed technique for mRNA–LNP production.

Finally, there is also the polymer–lipid hybrid vector approach that combines both of the above methods, offering stability and robust thermodynamic properties with efficient delivery of mRNA in the target [93].

## 4. Critical Quality Attributes for mRNA-Based Protein Replacement Therapies

Quality analysis is an integral part of the manufacture of every pharmaceutical product, as it guarantees the uniformity between different batches, its safety and its efficiency. The worldwide need for an encapsulated mRNA product as exemplified by mRNA–LNP COVID-19 vaccines, which has brought forth the need for the establishment of critical quality attributes (CQAs) of the product that also pertain to encapsulated mRNA designated for protein replacement therapies [42,94]. Being a novel and unique product both in the sense of the therapeutic agent itself and its pharmacotechnical characteristics, its CQAs are still being developed and standardized.

The current CQAs can be broken down into three main areas: those relevant to the mRNA molecule, those pertaining to the delivery system and finally those that characterize mRNA–LNP as a whole.

Concerning mRNA itself, before proceeding to encapsulation, the IVT product must be free of dsRNA, truncated species as well as DNA–RNA hybrids resulting from DNA digestion remnants [95]. Such contaminations could not only affect the efficiency of the product by activating intracellular detection mechanisms of foreign RNAs but could also activate cellular apoptosis or even increase the odds of triggering a cytokine storm in the recipient [72,73,96,97]. Furthermore, alongside with the 5′-capping efficiency and Poly-A tail length, the sequence and the quantity of the produced mRNA should be verified [42,94,98].

Delivery carrier CQAs include an exhaustive characterization of the lipids utilized in their formation, validating their identity and ratios, the absence of impurities, their distribution, electric charges and isoelectric points as well as their micromorphology and transfection efficiency [13,49].

Finally, the mRNA–LNP product must be characterized as well in terms of its particle size, zeta potential, release attributes and encapsulation efficiency [13,36]. Following the paradigm set by mRNA–LNP COVID-19 vaccines’ sensitivity toward storage conditions can be an issue and as such is included in the rational design of CQAs for encapsulated mRNA products [99,100]. Figure 5 summarizes the CQAs described above.

## 5. mRNA-Based Protein Replacement Therapies in Preclinical and Clinical Stage

A variety of diseases, some of which are—up to this day—untreated, can potentially be treated via mRNA-based protein replacement therapies. Such therapeutic approaches have been tested, both in preclinical and clinical studies, and some interesting data have emerged. Below, we present some mRNA replacement therapy cases, mostly referring to mice models of monogenic disorders.

Phenylketonuria (PKU) is an autosomal recessive disease caused by deficiencies in phenylalanine metabolism, due to mutations in the phenylalanine hydroxylase (PAH) gene [101]. The PAH gene encodes for a hepatic enzyme, responsible for breaking down phenylalanine into tyrosine. Its deficiency leads to toxically elevated accumulation of phenylalanine in plasma and/or organs such as the brain, thus resulting in irreversible intellectual disabilities [101]. PKU patients have to adhere to a strict diet, in conjunction with the administration of medication [102]. Of note, apart from the fact that it is difficult for people who eat high-protein meals to adhere to a strict diet, medications used to confront the disease are associated with inconvenient dosing and side effects [103,104]. Thus, alternative therapeutic approaches are needed in order to bring maximum benefits to the patients.

To this end, a study was conducted to develop an mRNA replacement therapy for PKU in mice models of the disease [105]. In more detail, a full-length mRNA encoding for human PAH was encapsulated in a lipid nanoparticle (LNP) and was delivered to a mouse model, carrying a missense mutation in the PAH gene. As a result, high levels of human PAH protein were generated in the hepatocytes, thus restoring phenylalanine metabolism. Interestingly, the reduction in phenylalanine did not coincide with any adverse clinical signs, either after single or repeat doses of therapy (in both males and females) [105]. Indubitably, these data establish a proof of principle of an mRNA-based protein replacement therapy to treat PKU.

In a similar manner, another study aimed to examine the ability of an LNP-encapsulated ARG1 mRNA—capable of translating into arginase, an enzyme catalyzing the final step in the urea cycle, thus converting ammonia to urea for excretion—to treat a conditional murine model of ARG1 deficiency [106]. Arginase deficiency (AD) is another autosomal recessive metabolic disorder caused by mutations in the ARG1 gene [107]. Patients suffer from progressive loss of psychomotor functions, spastic tetraplegia, hyperactivity of tendon reflexes, seizures, growth retardation and, in some cases, severe and/or lethal hepatic diseases, as urea cycle mainly occurs in the liver [108,109]. Treatments, once again, include protein-restricted diet, administration of nitrogen scavengers to reduce arginine and guanidino compounds levels and—when necessary—liver transplantation [110]. The results derived from Truong et al.’s study indicate that repetitive LNP–ARG1 treatment of AD mice prevents weight loss, corrects biomarkers of the disease without signs of hepatic toxicity and, most importantly, rescues from lethality [106]. As far as the latter is concerned, only the mice treated every 3 days survived for 77 days (the length of the study), whereas the mice treated weekly survived for 62 days [106]. These results further lend hope that patients with genetic diseases, as well as their physicians, will have another molecular therapy available for them in the future.

Another metabolic disease that has been found to be eligible for mRNA therapy is Fabry disease, a lysosomal disorder attributed to the deficiency of α-galactosidase A that is manifested with cardiomyopathy and renal failure [13]. In vivo studies conducted on mice and non-human primates demonstrated that the use of mRNA encoding α-galactosidase A successfully managed to improve the outcome of this disease [111]. Similarly, preclinical studies have reported salutary findings regarding the use of mRNA protein replacement therapy on other metabolic diseases still lacking effective treatment, such as Crigler–Najjar syndrome, hepatorenal tyrosinemia and acute intermittent porphyria [112,113].

More cases of transcript-based therapies, for certain liver diseases, have been presented in a review article [112]. In general, the preclinical studies discussed in that review refer to LNP-encapsulated mRNA molecules that translate into specific proteins (depending on the disease), resulting this way in improved health status of the mice receiving the mRNA. The authors, despite highlighting some technical limitations of mRNA replacement therapy, such as the delivery of the molecule to non-target tissues, conclude that one can be optimistic about hepatic transcript therapies reaching the clinic and healing patients in the future [112].

In addition, in another review article, the use of modified RNA in cardiac therapy is discussed [14]. The authors, more specifically, present preclinical studies in animal models of heart disorders (such as mice, pigs and monkeys of ischemic heart failure), in which RNA delivery was conducted—in terms of protein replacement therapy—in order to evaluate the ability of an mRNA to serve as a therapeutic tool for cardiac vascularization and regeneration. The authors conclude that such technology, under certain circumstances (that refer to increased safety and scalability, development of cost-effective clinical-grade materials, robust delivery methods and lower costs), may turn into an excellent therapeutic agents to induce cardiac regeneration and promote cardiac function [14]. Apart from cardiovascular disorders, blood disorders have also been targeted for protein replacement therapy through mRNA delivery. Such applications of mRNA-based protein replacement therapy include hemophilia A and B, namely bleeding disorders deriving from the deficiency of coagulation factors VIII and IX, respectively. The use of mRNA molecules encoding variants of these factors has been tested in vivo in mouse models of hemophilia A and B, displaying high efficacy in stimulating robust and durable expression of the corresponding missing factors [114,115].

In addition to all the aforementioned therapeutic approaches, mRNA replacement therapy was also positively assessed in preclinical studies for cystic fibrosis treatment [116,117]. Cystic fibrosis is primarily considered as a lung disease, caused by autosomal recessive mutations in the gene encoding for cystic fibrosis transmembrane conductance regulator (CFTR), a protein channel that controls the flow of H_2_O and CI- ions in and out of the cells inside the lungs. When CFTR is mutated, and thus works incorrectly, these ions cannot flow out of the cell, a condition leading to the buildup of thick mucus in the lungs, which in its turn leads to serious lung dysfunction [118]. The delivery of CFTR mRNA to mice lacking CFTR led to improvements in lung functional parameters, in a manner resembling FDA-approved cystic fibrosis drugs [116,117,119]. However, in order for such therapies to be clinically relevant, CFTR protein expression after mRNA delivery requires to be long-lived, as the frequent re-administration of the drug could probably lead to toxicities from high doses of the delivery vehicle [119].

As far as clinical trials are concerned, several clinical studies have been initiated in order to demonstrate the potential of an mRNA protein replacement therapy. It is worth noting that most of the clinical trial mRNA candidates use lipid nanoparticles in their formulation, indicating the importance of an adequate vehicle [120].

Translate Bio, a biotechnology company, initiated a clinical trial (which is currently ongoing) for the treatment of cystic fibrosis. They developed a product composed of an mRNA, encapsulated in a lipid nanoparticle, encoding for CFTR. The participants enrolled were 12 patients with cystic fibrosis receiving a single-ascending dose, via a nebulizer, of either mRNA or placebo. The results indicate that the drug was generally well tolerated in low and middle doses, while the patient’s lung function measurement increased [120]. Of note, it has been estimated that restoring 5% of wild-type CFTR mRNA in the cytosol is enough to ameliorate cystic fibrosis symptoms, although a higher threshold is necessary to avoid complications later in someone’s life [121].

The same company also initiated a clinical trial for the treatment of Ornithine Transcarbamylase Deficiency (OTC), using an OTC-encoding mRNA in a lipid nanoparticle. Unfortunately, the clinical trial was discontinued due to undesired data for the safety and pharmacokinetic profile of the drug [120]. The investors relate these data to the non-optimal features of the LNPs used, thus highlighting the importance of developing novel next-generation LNP vehicles [120].

Apart from Translate Bio’s clinical studies, the company Moderna, in collaboration with AstraZeneca, initiated two clinical trials using naked mRNA for the treatment of ulcers in diabetic II patients and heart failure patients [120]. Furthermore, Moderna also initiated two clinical trials to evaluate the safety and pharmacokinetic/pharmacodynamic profile of an mRNA drug, for the treatment of two more autosomal recessive diseases, Propionic Acidemia and Isolated Methylmalonic Acidemia, respectively [120]. Proper mRNAs encoding for the therapeutic enzymes, encapsulated in an LNP, have been used [120]. All these clinical trials are currently ongoing; thus, the results are yet to be published.

Remarkably, another mRNA-based protein replacement therapy that has passed the stage of preclinical testing and entered clinical development is a drug encoding vascular endothelial growth factor A (VEGF-A), an angiogenic factor that stimulates post-ischemic myocardium regeneration and boosts blood vessel growth [122]. In the ongoing EPICCURE phase 2a clinical trial sponsored by AstraZeneca, this therapy (AZD8601) is currently tested on patients with moderately compromised left ventricular function that undergo surgical revascularization [123]. Earlier, the outcomes obtained by the in vivo testing of AZD8601 were very promising, as this drug was not only reported to enhance blood flow and increase blood vessel density in the skin and heart of animal models, but also managed to boost the cardiac function of pigs undergoing experimental myocardial infarction [124,125]. Table 1 presents the clinical trials on mRNA protein replacement initiated so far.

## 6. Conclusions and Future Applications

Taken together, the latest preclinical findings and emerging clinical data increasingly suggest that mRNA therapy constitutes a very promising approach with the potential to complement the deficiencies of conventional therapeutic methods and even partially replace them. This auspicious prospect stems from the capacity of mRNA therapy to circumvent limitations associated with therapies employing DNA or recombinant protein technologies, offering the benefit of transient protein expression without the risk of genomic integration [5]. Despite these assets, this rapidly expanding technology initially displayed some limitations that have so far severely confined its applications on chronic medical conditions. These included the aforementioned transient protein expression that may also constitute a drawback, in tandem with the immunogenic properties of the mRNA molecule and the lack of well characterized and efficient delivery systems specifically to the target tissue [12].

While worldwide use of mRNA for vaccination purposes has been an exemplary practical application highlighting the viability of the mRNA–LNP technology, it must be stressed that using mRNA for protein replacement therapies may have additional peculiarities. It can be argued that although mRNA–LNPs used for protein replacement therapies are addressed to a smaller population than mRNA–LNPs used for immunization, they may require bigger doses and a more frequent administration schedule. For example, the mRNA-1273 COVID-19 vaccine is used in two 100 µg doses, with 25 μg doses also being effective, while trials with VEGF-A coding AZD8601 utilize thirty 100 μg to 1 mg doses and CFTR coding MRT5005 is being tested at 8 to 24 mg doses [123,131,132]. Given that mRNA vaccines present storage issues with regard to temperature stability, requiring dry ice temperatures for transportation and storage, the frequent and big dosage schemes of the latter are expected to impose considerable difficulties in their mainstream implementation as therapeutical products [133]. The storage condition problem can possibly be tackled by the utilization of techniques such as lyophilization, which can provide a product that retains its stability profile for twenty-five weeks at 4 °C and twelve weeks at room temperature (25 °C) [134,135,136]. To address the problem of mRNA protein replacement therapies requiring doses almost an order of magnitude larger than the ones used for vaccinations, companies can switch over to newer approaches, which can produce mRNA yielding more protein per coding molecule. The employment of self-amplifying mRNA, CircRNA and Endless RNA could potentially allow for lower mRNA requirements per dose, increasing the industry’s production output per year to meet patient demands [137,138]. Another consideration could be the size of the protein that needs to be coded by mRNA. Full S spike protein coded by the mRNA vaccines is about 180 kDa, whereas the mRNAs used in protein replacement therapy so far code for proteins in the range of 27–127 kDa as seen in Table 1 [139,140]. Should the need arise for longer mRNAs, delivery vector design can be revisited to be able to accommodate bigger constructs and deliver them efficiently. Such a need could also impact other integral parts of mRNA manufacture, such as purification and filtration, because longer mRNAs are more prone to shear damage (DAVIS). In this case, state-of-the-art purification and filtration techniques such as monolith chromatography and specialized tangential flow filtration membranes could be employed [141,142,143].

Although not all limitations impeding the use of this biomolecule in clinical practice have been resolved, the latest advances in the field of biomolecular engineering have eliminated significant barriers, paving the way for a more generalized use of this technology. This progress can be attributed to our deeper knowledge of this biomolecule, its enhanced stability through modifications of its structure and the development of novel delivery systems with enhanced safety and transfection efficacy. Despite the vast potential of the mRNA technology not only to revolutionize former therapeutic and diagnostic practices, but also to amend the rules of the pharmaceutic market through its cost-effective and widespread development, its applications are still mainly at the stage of assessment through clinical trials that are currently underway [13]. Although the long-term performance of mRNA therapeutics in real-world conditions and large-scale populations is yet to be seen, preliminary evidence remains very promising.

## Figures and Tables

**Figure 1 pharmaceutics-15-00166-f001:**
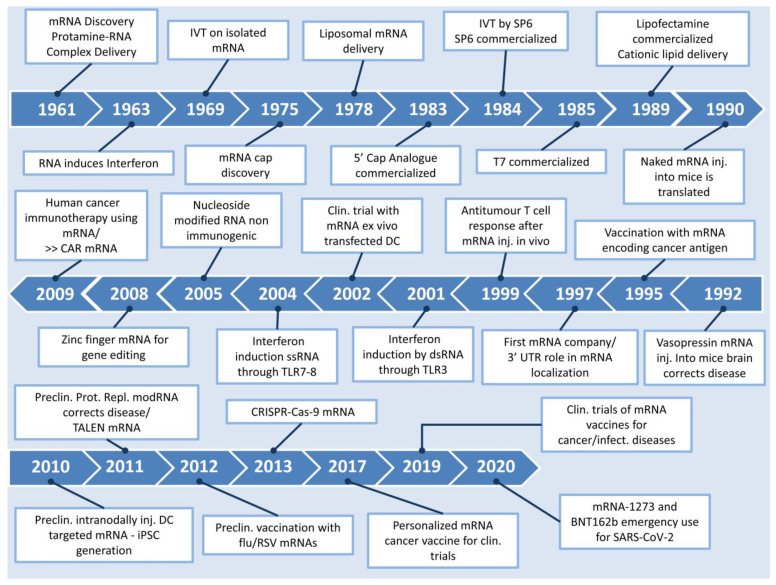
Timeline of major events in the history of mRNA. Clin: clinical, Preclin: preclinical, iPSC: induced pluripotent stem cells, Inj: injected, CAR: chimeric antigen receptor, DC: dendritic cells [8,9,10,19,20,21,22,23,24,25,26,27,28,29,30,31,32,33,34,35,36,37].

**Figure 2 pharmaceutics-15-00166-f002:**
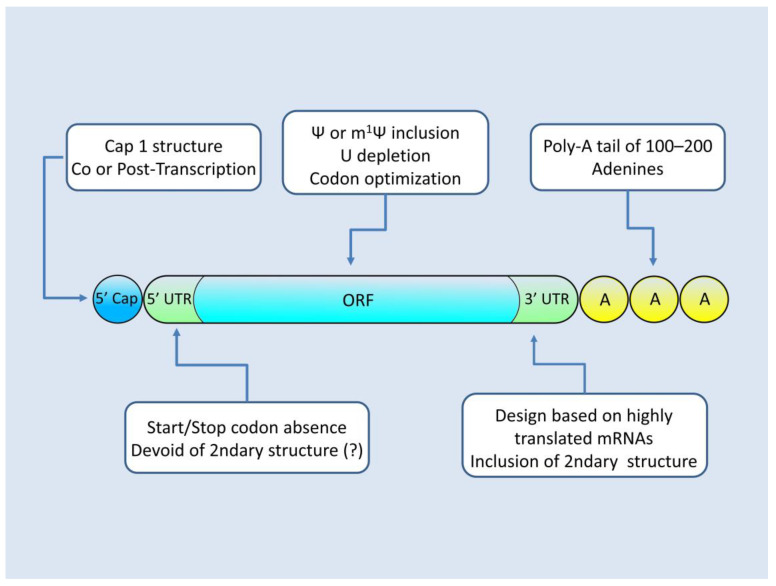
Architecture of modRNA. Modifications to the sequence (plasmid level) and to the chemistry (IVT level) allow for mRNAs with increased translational efficiency and minimal activation of the cell’s innate immunity mechanisms.

**Figure 3 pharmaceutics-15-00166-f003:**
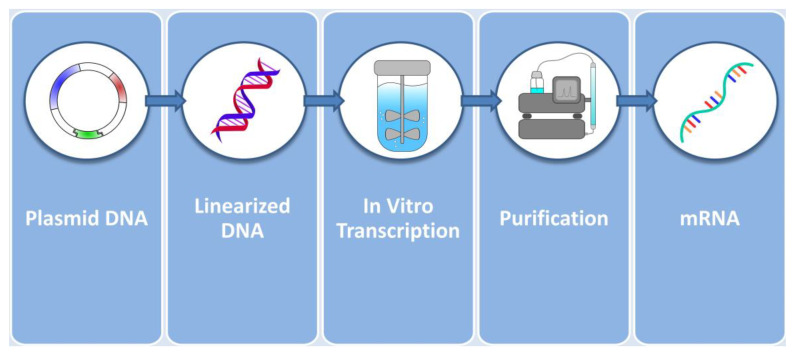
In vitro production of mRNA from plasmids. Plasmids coding for the intron-depleted optimized mRNA sequence are first linearized and then added to the bioreactor alongside with RNA polymerase, dNTPs, a buffer system and supplementary reagents. The resulting product is then purified by chromatography and flow filtration techniques and is ready to be encapsulated into delivery vectors.

**Figure 4 pharmaceutics-15-00166-f004:**
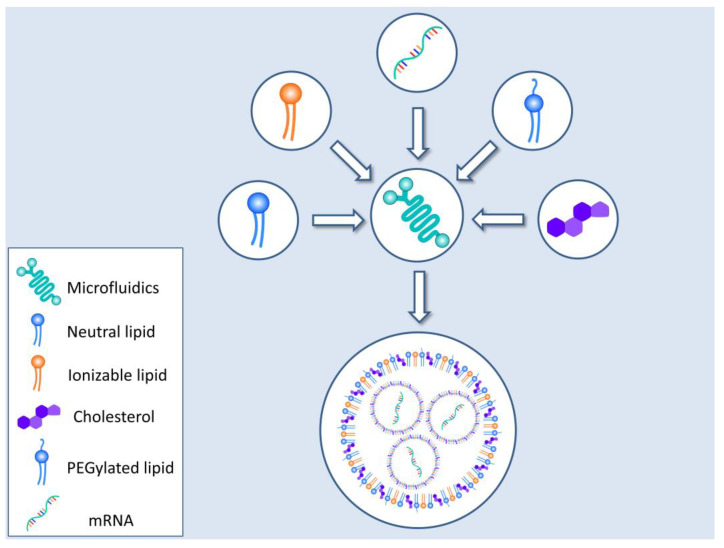
Lipid nanoparticle formation. LNPs are nowadays made by utilizing microfluidic technology which yields a uniform and well characterized final product. The encapsulation material (various natural or modified lipids) is fed into the device alongside with the mRNA. The resulting mRNA–LNPs are further purified to ensure suitability for human use.

**Figure 5 pharmaceutics-15-00166-f005:**
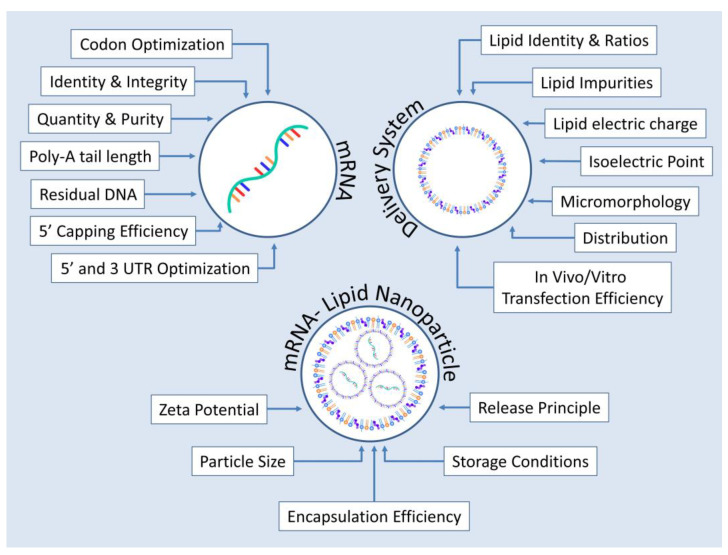
Critical quality attributes in the production of encapsulated mRNA products to be used in humans for protein replacement therapies. Qualities examined are the ones pertaining to the mRNA molecule encoding the protein of interest, to the delivery system as well as to the finished product which in this case is an mRNA–LNP formulation.

**Table 1 pharmaceutics-15-00166-t001:** Clinical trials that utilize mRNA for protein replacement therapies.

NCT NUMBER/PHASE	CONDITION	DELIVERY SYSTEM	ENCODING SEQUENCE/PROTEIN MOLECULAR WEIGHT	SUBJECTS	INTERVENTION	STATUS
NCT03370887/Phase II	Heart failure	Naked mRNA	VEGF-A 27 kDa [126]	Twenty-four patients with compromised left ventricular function that undergo surgical revascularization.	Patients had received either AZD8601 or placebo as epicardial injections and were followed up for six months.	Completed
NCT02935712/Phase I	Male subjects with type II diabetes	Naked mRNA	VEGF-A 27 kDa [126]	Up to sixty male patients with type II diabetes, aged 18–65 years old	In Part A, subjects had received an intradermal injection (ID) of either AZD8601 or placebo in a single ascending dose. In Part B, patients had received an ID injection of either AZD8601, in forearm skin, or the placebo.	Completed
NCT04159103/Phase I/II	Propionic Acidemia (PA)	LNPs	Alpha and Beta subunits of propionyl-CoA carboxylaseAlpha chain: 72 kDa Beta chain: 56 kDA [127]	Thirty-six patients with genetically confirmed PA, from one year old and older.	In Phase I, the patients will receive doses of mRNA-3927, for the dose optimization stage and subsequently for the dose expansion stage. In Phase II, the patients will receive the identified intravenous dose of mRNA-3927 and will be followed up for two years.	Recruiting
NCT03810690/Phase I/II	Methylmalonic acidemia (MMA)	LNPs	Methylmalonyl-coenzyme A mutase (MUT) 78 kDa [128]	Patients with methylmalonic academia, aged 1–18 years old, with elevated plasma methylmalonic acid.	The patients were about to receive doses of mRNA-3704, for the dose escalation phase, and subsequently for the dose expansion stage.	Withdrawn
NCT03767270/Phase I/II	Ornithine transcarbamylase deficiency (OTCD)	LNPs	Ornithine transcarbamylase 36.1 kDa [129]	Subjects with OTC Deficiency	The patients were about to receive intravenous, single-ascending low, mid and high doses of MRT5201 or the placebo.	Withdrawn
NCT03375047/Phase I/II	Cystic fibrosis (CF)	LNPs	Human cystic fibrosis transmembrane regulator protein (CFTR) 127 kDa [130]	Forty adult subjects with CF	The patients are supposed to receive single and multiple escalating doses of MRT5005, administered by nebulization to the respiratory tract, or the placebo.	Unknown

## Data Availability

Not applicable.

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
