# Peer review of "mRNA in the Context of Protein Replacement Therapy"

_pharmaceutics, 2023, doi:10.3390/pharmaceutics15010166_

Round 1

Reviewer 1 Report

Overall, it is a good review paper on mRNA vaccine or Protein Replacement Therapy. I have some minor suggestions for the authors.

1, Ln 55 - Ln 62, There are sentences describe the mechinism of miRNAs and siRNAs without references. I suggest to include a reference that summarizing the mechinism or the first paper found the mechinism.

2, Ln 100, Lack the reference for the fact.

3, Ln 200, missing space between "aminoacid", should be amino acid.

4, Ln 253, This paragraph describe the advantages and disadvantages of the mRNA delivery methods. I suggest to include a table for that, which will be more clear and concise.

5, I noticed the authors use two reference styles, "Author, year" and ref number [xx].  e.g. Ln 409, Ln 402

Author Response

Reviewer #1

Overall, it is a good review paper on mRNA vaccine or Protein Replacement Therapy. I have some minor suggestions for the authors.

We would like to thank the reviewer for carefully reading the manuscript and providing us with valuable suggestions to enhance it. Please find a numbered list of replies to your comments.

Comment 1. Ln 55 - Ln 62, There are sentences describe the mechinism of miRNAs and siRNAs without references. I suggest to include a reference that summarizing the mechinism or the first paper found the mechinism.

Reply to the comment:

We agree with the reviewer’s accurate remark and selected to cite a paper thoroughly analyzing the aforementioned mechanisms [1].

In the manuscript:

We added a reference in line 59.

Comment 2. Ln 100, Lack the reference for the fact.

Reply to the comment:

We agree with the reviewer’s comment and proceeded with the addition of a reference in line with our respective points [2].

In the manuscript:

We added a reference in line 101.

Comment 3. Ln 200, missing space between "aminoacid", should be amino acid.

Reply to the comment:

We detected this spelling issue and made the necessary changes in the manuscript.

In the manuscript: 

This word was corrected in line 200.

Comment 4. Ln 253, This paragraph describe the advantages and disadvantages of the mRNA delivery methods. I suggest to include a table for that, which will be more clear and concise.

Reply to the comment:

Although we acknowledge the reviewer’s accurate remark, we did not opt for a table format to display this section of our paper as this is not one of the main focal points of our study. Consistent with the above, the data included in this rough comparison of the various modalities of mRNA delivery are not the outcome of an extensive scrutiny, but rather an attempt to include a brief comment on this matter, and therefore do not constitute eligible material for the recommended layout.

In the manuscript:

We did not make any changes corresponding to this point.

Comment 5. I noticed the authors use two reference styles, "Author, year" and ref number [xx].  e.g. Ln 409, Ln 402

Reply to the comment:

We thank the reviewer for these accurate remarks and we proceeded to all the necessary changes, choosing to maintain the style of reference number.

In the manuscript:

We made changes in the manner that references are cited in lines 366, 375, 402 and 409.

 [1]         Damase TR, Sukhovershin R, Boada C, Taraballi F, Pettigrew RI, Cooke JP. The Limitless Future of RNA Therapeutics. Front Bioeng Biotechnol 2021;9:628137. https://doi.org/10.3389/fbioe.2021.628137.

[2]          Qin S, Tang X, Chen Y, Chen K, Fan N, Xiao W, et al. mRNA-based therapeutics: powerful and versatile tools to combat diseases. Signal Transduct Target Ther 2022;7:166. https://doi.org/10.1038/s41392-022-01007-w.

Reviewer 2 Report

The authors provided a systematic and concise review of utilizing mRNA for the protein replacement therapy. However, there are several major recommendations needed to be addressed before the final acceptance of the paper. 

1. Please provide the schematic figure of LNP-based delivery methods of mRNA and listed all the components discussed in the encapsulation session. Do the PEG, cholesterol or other DODMA or DODAP affect the aggregation or other properties of LNP which were also discussed in many published review papers such as Hou, X., Zaks, T., Langer, R. et al. Lipid nanoparticles for mRNA delivery. Nat Rev Mater 6, 1078–1094 (2021). Please provide some more detailed discussion about it. 

2. What are the significant differences in the requirement in mRNA-based protein replacement therapy and mRNA-based vaccine? Protein size? Loading efficiency? Considering mRNA provides a transient protein replacement strategy, what are the major benefits of mRNA compared to utilize genome editing technology to insert the functional DNA sequences to the safe harbor site in the human genome?

3. In the manuscript, the authors briefly discussed the synthetic materials used in the delivery of mRNA. It is not very helpful to understand the big picture of using or constructing different materials in the delivery of mRNA, Therefore, it would be easier for readers to comprehend the advanced of material-wide mRNA delivery by reviewing the use of different materials in different diseases or different administration strategies of mRNA-loaded vehicle in different diseases which may illuminate the more efficient development of mRNA-based protein replacement therapy.

4. It is highly recommended to list the molecular weight of each protein in the Table 1 and discuss about how the size of mRNA or protein-of-interest will affect the final therapeutic outcomes. 

5. What is the population size or percentage that needs protein replacement therapy? Will the storage of mRNA and stability of mRNA-based therapy hurdle the value of such therapeutic strategy? Please also provide a brief and concise discussion regarding those issues. 

Author Response

Reviewer # 2

The authors provided a systematic and concise review of utilizing mRNA for the protein replacement therapy. However, there are several major recommendations needed to be addressed before the final acceptance of the paper.

We would like to thank the reviewer for carefully reading our manuscript and making such an accurate remark to further improve it.

Comment 1. Please provide the schematic figure of LNP-based delivery methods of mRNA and listed all the components discussed in the encapsulation session. Do the PEG, cholesterol or other DODMA or DODAP affect the aggregation or other properties of LNP which were also discussed in many published review papers such as Hou, X., Zaks, T., Langer, R. et al. Lipid nanoparticles for mRNA delivery. Nat Rev Mater 6, 1078–1094 (2021). Please provide some more detailed discussion about it.

Reply to the comment:

Figure 4 provides a basic schematic of the LNP-based delivery vector and how it is created. We opted to incorporate only the basic elements needed for the formation of the vector, compared to the more exhaustive approach proposed, in order to allow the reader to better focus on the core concept of mRNA-LNPs. Concerning the comment on component’s effect on LNP characteristics, we have enriched section 3 with some specifics to give the reader a better understanding of the matter at hand. Substantiating the claims in this section has also allowed us to showcase literature more geared towards answering the questions posed in comment 1, should the reader desire to learn more.

In the manuscript:

Please see the addition at section 3, lines 297-314: “It has to be noted that critical properties of the LNPs, dictating their behavior and stability [...] have the propensity to accumulate in the spleen, hence they could be of use when targeting this organ”.

Comment 2. What are the significant differences in the requirement in mRNA-based protein replacement therapy and mRNA-based vaccine? Protein size? Loading efficiency? Considering mRNA provides a transient protein replacement strategy, what are the major benefits of mRNA compared to utilize genome editing technology to insert the functional DNA sequences to the safe harbor site in the human genome?

Reply to the comment:

The major benefits of mRNA compared to approaches intervening on the genome level have been included in the original version of our manuscript both in the introduction and the discussion. Using your comment of the comparison between the differences in mRNA utilization in protein therapy and vaccination as a steppingstone, we added a second paragraph on our conclusions and future applications section. This paragraph, apart from highlighting the major differences, has given us the opportunity to furtherly expand on issues raised in comment 4 and comment 5.

In the manuscript: 

Please see section 6, lines 519-549: “While worldwide use of mRNA for vaccination purposes… such as Monolith chromatography and specialized tangential flow filtration membranes could be employed”

Comment 3. In the manuscript, the authors briefly discussed the synthetic materials used in the delivery of mRNA. It is not very helpful to understand the big picture of using or constructing different materials in the delivery of mRNA, Therefore, it would be easier for readers to comprehend the advanced of material-wide mRNA delivery by reviewing the use of different materials in different diseases or different administration strategies of mRNA-loaded vehicle in different diseases which may illuminate the more efficient development of mRNA-based protein replacement therapy.

Reply to the comment:

Although we share the reviewer’s interest in the concept of mRNA delivery, a more extended analysis of the various materials involved in this technology would exceed the purposes of this review. As implied from the title, our main focal point was protein replacement therapy and therefore the whole study is oriented towards this field combining a clinical and biotechnological point of view. Such an approach as the one recommended would be very technical and prone to radically change the scope of our study.

In the manuscript: 

We did not make any changes corresponding to this point.

Comment 4. It is highly recommended to list the molecular weight of each protein in the Table 1 and discuss about how the size of mRNA or protein-of-interest will affect the final therapeutic outcomes.

Reply to the comment:

The molecular weights have been added as suggested in Table 1. Furthermore section 6 now includes relevant commentary on the variability of mRNA sizes that might be encountered when designing a protein replacement therapy mRNA and potential documented issues.

In the manuscript: 

Table now includes the molecular weights of each protein mentioned.

Please also see section 6, lines 519-549: “While worldwide use of mRNA for vaccination purposes… such as Monolith chromatography and specialized tangential flow filtration mem-branes could be employed”

Comment 5. What is the population size or percentage that needs protein replacement therapy? Will the storage of mRNA and stability of mRNA-based therapy hurdle the value of such therapeutic strategy? Please also provide a brief and concise discussion regarding those issues.

Reply to the comment:

Protein replacement therapy using mRNA, as it stands, targets those groups of patients that suffer from usually rare monogenic diseases. Although their numbers might be lower than the number of people requiring an mRNA vaccine during the pandemic, the more frequent dosing of the former group combined with the generally larger doses needed for protein therapy raises some serious issues. These pertain, of course, to the storage and stability of the product as you aptly remarked, as well as to production issues to meet the demand. We have enriched section 6 (second paragraph) in a way that reflects those issues and goes a step further to propose some viable solutions to those issues.

In the manuscript: 

Please see section 6, lines 519-549: “While worldwide use of mRNA for vaccination purposes… such as Monolith chromatography and specialized tangential flow filtration membranes could be employed”.

Reviewer 3 Report

The review report was good. Some shortcomings are written in the attached file

Author Response

Reviewer # 3

In this review, Vavilis et al., have discussed importance and application of mRNA as a tool for protein replacement therapy in various human diseases. The author also presented information pertaining to the manufacture of modified mRNA through in vitro transcription, delivery vehicles and critical quality attributes pertaining to the finished products. Although the content of this article is descriptive but there are several points to be addresses to enrich it further. There are some grammatical mistakes which need to be corrected as well.

We would like to thank the reviewer for carefully reading our manuscript and providing insightful feedback that enables us to improve our manuscript.

Comment 1. Title: the word ‘service’ is not suitable here, please rephrase the title

Reply to the comment:

We have rephrased the title to “mRNA in the context of Protein Replacement Therapy”

In the manuscript:

Title has been changed to “mRNA in the context of Protein Replacement Therapy”

Comment 2. No reference mentioned from line 86 to 122?

Reply to the comment:

Taking into consideration the reviewer’s useful comment, we cited several studies to substantiate statements made in the relevant text.

In the manuscript: 

We added three references in lines 88, 102 and 117, respectively.

Comment 3. Line 114, ‘Oppositely, non-viral delivery, as in the form of plasmid DNA’…. Plasmid DNA is not a non-viral delivery vector. Non-viral delivery platforms like liposome, dendrimers, solid lipid nanoparticles, polymeric micelle, etc. can deliver plasmid DNA. Please rephrase the sentence. Again, the term ‘transduction’ used in this sentence is not correct. For non-viral delivery it should be ‘transfection’. Transfection is the process of introducing nucleic acids into cells by non-viral methods. And, ‘Transduction’ is the process whereby foreign DNA is introduced into another cell via a viral vector

Reply to the comment:

The comment has been taken into account and relevant changes have been made to the text.

In the manuscript: 

The passage has been changed to “Oppositely, non-viral delivery, as in the form of liposomes, dendrimers, solid lipid nanoparticles and polymeric micelles used to deliver plasmid DNA, has the benefit of lower immunogenic and oncogenic capacity, as well as high tissue specificity, yet dis-playing very poor transfection efficiency”.

Comment 4. In Figure 1, need to add reference, for each discovery mentioned in the timeline

Reply to the comment:

We have added all the appropriate references that constitute the major discoveries pictured in the diagram.

In the manuscript: 

Please see the legend of figure 1.

Comment 5. Size of Figure 2,3,4 seems very big

Reply to the comment:

The full size figures where incorporated in the text mainly as “placeholders” indicative of their final placement and as such should not be considered as being in their final size. The original figure files have been submitted to the editorial office alongside with the manuscript, so they can be adjusted accordingly should the article be published.

In the manuscript: 

We did not make any changes corresponding to this point.

Comment 6. Rephrase section ‘4. Critical Quality Attributes for protein replacement therapies mRNA’

Reply to the comment:

The section title has been rephrased to read “Critical Quality Attributes for mRNA-based protein replacement therapies”

In the manuscript: 

We changed the section title to “4. Critical Quality Attributes for mRNA-based protein replacement therapies”.

Comment 7. Although the authors entitled the review based on mRNA protein replacement therapy, they have included information more on generalized mRNA and it’s delivery rather than discussing broadly about it’s implication, challenges in protein replacement therapy

Reply to the comment:

We acknowledge the reviewer’s accurate remark on the extensive nature of our analysis. However, this being a review article, we deemed it necessary to provide readers with a more comprehensive overview of the topic by framing our study’s focal point with complementary sections integrating pertinent information to our study, thus helping readers better assimilate latest findings in the field.

In the manuscript: 

We did not make any changes corresponding to this point.

Comment 8. Line 31, 73 (italic, in vivo/in vitro) and elsewhere, Line 254, ‘Principal’, spelling check 

Reply to the comment:

We detected these spelling issues and made the necessary changes where applicable

In the manuscript: 

We corrected the italic formatting of in vivo/ in vitro throughout the manuscript and replaced the word principals with principles in line 255.